# Gender Differences in Acute Aortic Dissection

**DOI:** 10.3390/jpm12071148

**Published:** 2022-07-15

**Authors:** Eduardo Bossone, Andreina Carbone, Kim A. Eagle

**Affiliations:** 1Cardiac Rehabilitation Unit, Cardarelli Hospital, 9 Antonio Cardarelli Street, 80131 Naples, Italy; 2Unit of Cardiology, University of Campania “Luigi Vanvitelli”, 80138 Naples, Italy; andr.carbone@gmail.com; 3Cardiovascular Center, University of Michigan, Ann Arbor, MI 48109, USA; keagle@med.umich.edu

**Keywords:** gender, acute aortic dissection, pregnancy, type A aortic dissection, type B aortic dissection, sex

## Abstract

Cardiovascular disease (CVD) represents the most important cause of mortality and morbidity worldwide. There is heterogeneity in the epidemiology and management of CVD between male and female patients. In the specific case of acute aortic dissection (AAD), women, at the time of diagnosis, are older than men and complain less frequently of an abrupt onset of pain with delayed presentation to the emergency department. Furthermore, a history of hypertension and chronic obstructive pulmonary disease is more common among women. In type A AAD, women more often experience pleural effusion and coronary artery compromise, but experience less neurological and malperfusion symptoms. They undergo less frequent surgical treatment and have higher overall in-hospital mortality. Conversely, in type B AAD no significant differences were shown for in-hospital mortality between the two genders. However, it should be highlighted that further studies are needed in order to develop AAD gender specific preventive, diagnostic and therapeutic strategies.

## 1. Introduction

Cardiovascular disease (CVD) represents the most important cause of mortality and morbidity worldwide [1,2]. Interestingly, there is evidence of heterogeneity in the mechanism, risk factors, clinical characteristics, and the short- and long-term outcomes of CVD between women and men [3]. Furthermore, women are under-represented in cardiovascular clinical trials, which has reduced the elaboration of gender-specific approaches that could improve guideline recommendations and physicians’ adherence [4]. The aim of the present report was to highlight gender differences in patients with acute aortic dissection (AAD), including risk factors, presenting clinical features, diagnosis, management and outcomes. Pregnancy-related AAD is also discussed.

## 2. AAD

AAD represents a life-threatening emergency, needing prompt diagnosis and appropriate therapeutic interventions (Figure 1) [5,6]. The real incidence of AAD is arduous to define, due to pre-hospital mortality and/or missing diagnoses. Population-based studies show an incidence of 2.6 to 3.5 cases per 100,000 person-years (67% Type A; 33% Type B) [7,8]. Men are more affected than women (9.1 vs. 5.4 per 100,000 men and women, respectively; *p* < 0.001) [9,10]. Interestingly, women are aged more than men at the time of AAD onset [11]. Due to non-specific symptoms and signs, the diagnosis is often delayed [11]. Therefore, a high index of suspicion is crucial to make the diagnosis. In this regard a specific integrated diagnostic algorithm has been designed, including clinical risk assessment along with biomarkers (D-dimer) and imaging techniques (computed tomography being by far the most utilized) (Figure 2 and Figure 3) [12,13]. Urgent surgery is recommended for type A AAD, while medical treatment is usual for uncomplicated Type B AAD [5,6,14]. Thoracic endovascular aortic repair (TEVAR) is generally indicated in the case of complicated type B AAD [5]. The International Registry of Acute Aortic Dissection (IRAD) highlighted that in-hospital mortality for type A AAD has a downward trend from 31% to 22% (*p* < 0.001), over 17 years (December 1995 to February 2013), principally related to decreased surgical mortality (25% to 18%; *p* < 0.003) [15]. However, no changes in-hospital mortality rates for type B AAD (12% to 14%) were observed (Figure 4) [16,17]. Given that AAD is a life-long condition, involving the whole aortic system (holistic concept), patients should obtain optimal blood pressure (≤120/80 mmHg) and heart rate (≤60 bpm) control and imaging follow up (MRI or CT) in order to prevent aorta-related death and major complications [5,6,10,17].

## 3. Impact of Gender in AAD

### 3.1. Type A AAD 

In major population-based studies and registries, women were less frequently affected by type A AAD than men (31 to 48% vs. 52 to 69% respectively) [19,20,21,22,23,24,25]). Furthermore, at time of diagnosis, women were older than men (an average of 6–7 years older) [19,20,21,22,23,24,25].

In this regard, more than 50% of women experienced type A AAD at ≥70 years of age [19,20,21,22,23,24,25,26]. However, Liu et al. [27], in a Chinese population-based study (884 patients; 76.1% male, mean age 51.4 ± 11.8 years) did not show a significant gender difference in age (women 51.4 ± 11.8 vs. men 55.1 ± 12.5 years; *p* = 0.10).

The German Registry for Acute Aortic Dissection Type A (GERAADA; 56 centers; 3380 patients; 1234 (37%) women vs. 2146 (63%) men), showed that women were older than men (65.5 ± 12.7 vs. 59.2 ± 13.3 years; *p* < 0.001) [23]. The extension of AAD to the abdominal aorta was more frequent in men than in women (43% vs. 39%; *p* = 0.01) [23]. Visceral and renal malperfusion were more frequently diagnosed in men (women 32.8% vs. men 38.5%, *p* = 0.001) [23]. While aortic roots replacement was more frequent in men (21.6 vs. 17.7%; *p* < 0.001), distinct aortic-arch-repair approaches were distributed similarly in both genders (*p* = 0.094). Thirty-day mortality (women 16.3% vs. men 16.6%; *p* = 0.177), as well as the incidence of hemiplegia or hemiparesis after surgery (women 11.5% vs. men 10.1%; *p* = 0.240), did not differ between the two genders [23].

In the Nordic Consortium for Acute Type A Aortic Dissection (NORCAAD; 1154 patients with type A AAD surgical treatment; January 2005-December 2014) women represented 32%, and were significantly aged (65 ± 11 vs. 60 ± 12 years; *p* < 0.001), more often had hypertension (58 vs. 48%; *p* = 0.001) and chronic obstructive pulmonary disease (COPD) and had lower body mass index (26 ± 5 vs. 27 ± 4 kg/m^2^; *p* < 0.001), compared with men (8.3% vs. 4.9%; *p* = 0.03) [20]. Hypothermic cardiac arrest time and operation time were shorter among women (343 ± 133 vs. 374 ± 134.8 min; *p* < 0.001). There was no difference in intraoperative death or 30-day mortality between the genders (men 6.7% vs. women 9.1%; *p* = 0.17) [20].

In a recent metanalysis of nine studies comparing clinical outcomes according to gender in type A AAD patients treated surgically [28], women and men showed similar in-hospital/30-day mortality (RR = 1.04; 95% CI, 0.85–1.28; *p* = 0.67), risk of post-surgical stroke (RR = 1.07; 95% CI, 0.91–1.25; *p* = 0.43), and dialysis (RR = 0.84; 95% CI, 0.59–1.19; *p* = 0.32).On the other hand, a lower risk of reintervention for bleeding (RR = 0.84; 95% CI, 0.75–0.94; *p* < 0.01) was shown in women [28].

Table 1 and Table 2 showed clinical characteristics and outcomes of type A AAD of the major population based studied. 

### 3.2. Type B AAD 

Few studies have investigated the impact of gender in type B AAD [26,27,29,30]. Liang et al. [29] identified, from 2009 to 2012, 9855 (women 43.6%, n = 4293) patients with type B AAD. Women experienced AAD at a later age, had more comorbidities (heart failure, COPD, diabetes, rheumatologic disorders), and were more often medically managed compared to men (87.4 vs. 81.8%; *p* < 0.001) [29]. There were no differences in gender of unadjusted in-hospital mortality rates (women 11.6% vs. men 10.7%; *p* = 0.2) [29]. In an adjusted propensity-weighted regression analysis, gender did not significantly influence in-hospital mortality (OR = 0.91; 95% confidence interval [CI] 0.79–1.00; *p* =0 0.2) or stroke rates (OR = 0.91; 95% CI 0.51–1.57; *p* = 0.7), but women were less likely to have acute renal injury during hospitalization (OR = 0.68; 95% CI 0.60–0.70; *p* < 0.001) and more likely to experience cardiac events when undergoing open repair (OR = 1.45; 95% CI 1.01–2.11; *p* = 0.04) [29]. In particular, elderly women (>70 years old) experienced less acute renal failure (OR = 0.72; 95%CI 0.62–0.85; *p* < 0.001) but had higher odds of acute cardiac events compared to elderly men (OR = 1.20; 95% CI 1.04–1.50; *p* = 0.02) [29].

Takahashi et al. [30] analyzed data about 2372 (695 women, 29.3%) patients with type B AAD, enrolled in the Tokyo Acute Super-Network Registry. Women were older than men and presented later at the emergency department. Women showed a higher proportion of intramural hematoma (63.7% vs. 53.7%, *p* < 0.001) and were more medically treated (90.9% vs. 86.3%, *p* = 0.002), with higher in-hospital mortality (5.3% vs. 2.7%, *p* = 0.002) [30]. At multivariate analysis, female gender was not associated with higher in-hospital mortality (OR 1.67 [95% CI, 0.96–2.91]) [30].

Table 1 and Table 2 summarized clinical characteristics and outcomes of type B AAD of the major population based studied. 

## 4. Insights into Gender Related AAD from the IRAD 

The IRAD, established in 1996 at the University of Michigan, Ann Arbor, USA, is an observational registry involving 53 highly specialized aortic centers around the world, aiming to assess diagnoses, management, and outcomes of AAD [15,16,31].

According to IRAD data [11] (Table 3), AAD was more frequent in men but women were generally older (overall population n = 1078, 32% women; 49.7% of women were 70 years of age or older vs. 28.6% of men). The type A/B AAD ratio was approximately 2:1 in both genders [11]. Previous cardiac surgery was more common in men, while hypertension was more prevalent in women [11]. Other etiologies or risk factors (atherosclerosis, diabetes, Marfan syndrome, bicuspid aortic valve, cocaine abuse, iatrogenic dissection, previous aortic dissection, previous aortic aneurysm,) were similar between genders [11].

The presentation to hospital, after symptom onset, was significantly more delayed in women than in men (mean absolute difference of 4.7 h), negatively affecting outcomes [11]. Although typical presentation with chest pain was similar in women and men, women were less likely to report an abrupt onset of pain (*p* = 0.004) [11]. Congestive heart failure (*p* = 0.03) and coma/neurologic alterations (*p* = 0.05) were more common in women [11]. Electrocardiographic findings were similar in the two groups. CT was the most utilized imaging in both genders (>70%; 80.6% in men and 76.6% in women) [11]. Tomographic findings suggestive of periaortic hematoma (*p* = 0.03), pericardial effusion (38.8% vs. 28.6%; *p* = 0.001), pleural effusion (26.1% vs. 14.7%; *p* < 0.001) and coronary artery involvement (10.8% vs. 6.9%; *p* = 0.05), were more frequent among women [11]. Initial medical management with intravenous beta-blockers was less used in women than in men (62.1% in men vs. 55.6% in women; *p* = 0.05) [11].

There was no difference in surgical techniques for both type A and B AAD, however more women were medically treated than men (35% in men vs. 45.7% in women; *p* = 0.001) [11]. In-hospital complications, such as hypotension (0.001) and cardiac tamponade (*p* = 0.007) occurred more among women [11]. On the other hand, limb ischemia was more common in men (*p* = 0.04) [11].

Women showed higher overall in-hospital mortality (type A AAD + type B AAD) than men (*p* = 0.001) [11]. Women had lower survival than men for type A AAD (log rank *p* = 0.01) but not for type B AAD (log rank *p* = 0.47) [11]. Furthermore, women showed the greatest in-hospital mortality for surgically treated type A AAD (31.9% mortality in women vs. 21.9% in men, *p* = 0.013) [11].

No significant gender-related difference was shown for type A AAD medically treated mortality [11]. Interestingly, in the advanced age cohort (>75 years) women with type A AAD were treated more with only medical treatment than men (31.4% vs. 14.0%; *p* = 0.04). In the analysis stratified by age (age < 50, 50 to 65, 66 to 75, and >75 years), major differences in mortality between gender were shown in the 66- to 75-year age group (36% vs. 16%; *p* = 0.001) [11]. Older age and less typical symptoms at onset of AAD have been proposed as possible factors contributing to poorer outcomes in women.

A more recent analysis of an IRAD-Interventional Cohort (IVC) [32], consisting of more than 2823 type A AAD patients treated with endovascular, surgical, or hybrid procedures has partly confirmed the previous IRAD data about gender differences (Table 1). Of particular interest was the fact that, overall, in-hospital mortality was 16.7% in women (n = 162) and 13.8% in men (n = 256, *p* = 0.039). The frequency of postoperative complications was similar between genders, except for acute kidney injury, which was lower in women (17.7% vs. 21.2%, *p* = 0.029) [32].

However, five-years survival (82.6% in women vs. 85.9% in men) and reintervention (87.8% in women vs. 87.6% in men) were similar between genders [32]. Furthermore, the authors found that operative approaches were different between genders. As a note, complete arch replacement (OR = 7.3; 95% CI 2.07–25; *p* = 0.002) along with age (OR = 1.04; 95% CI 1.03–1.05; *p* < 0.001), renal failure (OR = 2.68; 95% CI 1.91–3.74; *p* < 0.001), coma (OR = 13.38; 95% CI 7.87–22.73; *p* < 0.001), limb ischemia (OR = 1.87; 95% CI 1.23–2.86; *p* = 0.003), and cardiopulmonary bypass time (OR = 1.01; 95% CI 1.01–1.01; *p* < 0.001) were independent predictors of in-hospital mortality in both genders [32].

In summary, data from IRAD highlighting women as compared to men show women to have the following: (a) older age, higher incidence of a history of hypertension and more delayed presentation to hospital; (b) more complications, such as periaortic hematoma, pericardial effusion, pleural effusion and coronary artery involvement; (c) higher overall and surgical type A AAD in-hospital mortality.

## 5. AAD in Pregnancy

Among 9707 AAD patients (3341 women [34.4%]) enrolled in the IRAD registry from 1998 to 2019, 29 women (0.3%, mean age, 32 ± 6 years) experienced pregnancy-related AAD, 13 (45%) had type A AAD and 16 (55%) had type B AAD [33,34,35,36]. AAD occurred in 15 pregnant women (4 in the first and 11 in the third trimester) and in 12 during the post-partum period. Twenty women (69%) had predisposing conditions: 13 (65%) Marfan syndrome, 2 (10%) Loeys-Dietz syndrome, 2 (10%) BAV, 2 (10%) family history of aortic disease and 1 (5%) familial thoracic aortic aneurysm [35]. In type A AAD, the mean aortic diameters at diagnosis were 54.5 (±5) at sinus of Valsalva and 54.7 (±7) at ascending aorta and in type B AAD was 32.5 (±5) at descending aorta [33]. Twenty-eight women (97%) survived at hospital discharge [mortality rate of 3% (1/29)] [33].

In The NHLBI National Registry of Genetically Triggered Thoracic Aortic Aneurysms and Cardiovascular Conditions (GenTAC) [37], among 94 women with Marfan syndrome, 10 (10.6%) experienced aortic complications during or post pregnancy (4 type A and 3 type BAAD). Regarding postpartum, 5 of 7 AAD including all 3 type B, experienced complications. Only 5 of 8 women were conscious of their Marfan syndrome diagnosis [37].

In the Nationwide Inpatient Sample of more than 10 million pregnancies and 41,000 AADs from 1998 to 2008, 44 cases of AAD in pregnancy were described, representing 0.1% of all cases of AAD. Only 7 of the 44 cases had Marfan syndrome, with 2 other women having other congenital anomalies [38]. The in-hospital mortality was of 6.8% for AAD during pregnancy vs. 15.4% of all AAD [38].

Kamel et al. [39] described 36 cases of pregnancy-related AAD, out of 6,566,826 pregnancies from 2005 to 2013. Aortic complications occurred in 5.5 (95% confidence interval [CI], 4.0–7.8) per million patients during pregnancy and in postpartum, compared with 1.4 (95% CI, 0.7–2.9) per million during the equivalent period 1 year later [38]. Pregnancy was associated with a significantly higher risk of AAD (incidence rate ratio, 4.0; 95% CI, 2.0–8.2) compared to the control period 1 year later [39].

Thus, AAD is a rare, but serious, complication of pregnancy [in the majority of cases type A AAD (57–80%)] [40,41,42,43]. There is no evidence of increased in-hospital mortality in pregnancy-related AAD. However, it should be underlined that the recognition of predisposing conditions (namely aortopathies, often not diagnosed until the acute event) and the monitoring of the aorta diameters throughout pregnancy and in post-partum, may reduce complication rates and improve outcomes.

## 6. Conclusions

A comprehensive understanding of gender differences in AAD is lacking. For both type A and B AAD, women, at the time of diagnosis, are older than men and complain less frequently about abrupt onset of pain, with delayed presentation to the emergency department. Furthermore, a history of hypertension and chronic obstructive pulmonary disease is more common among women. In type A AAD, women more often experience pleural effusion and coronary artery compromise but experience less neurological and malperfusion symptoms. They undergo less frequent surgical treatment and have higher overall in-hospital mortality. On the other hand, in type B AAD no significant differences are registered for in-hospital mortality between the two genders. A greater knowledge of gender differences in AAD risk factors, clinical presentation and treatment may improve diagnostic accuracy, along with short- and long-term prognosis. However, it should be highlighted that further studies are needed in order to develop AAD gender-tailored preventive, diagnostic and therapeutic strategies.

## Figures and Tables

**Figure 1 jpm-12-01148-f001:**
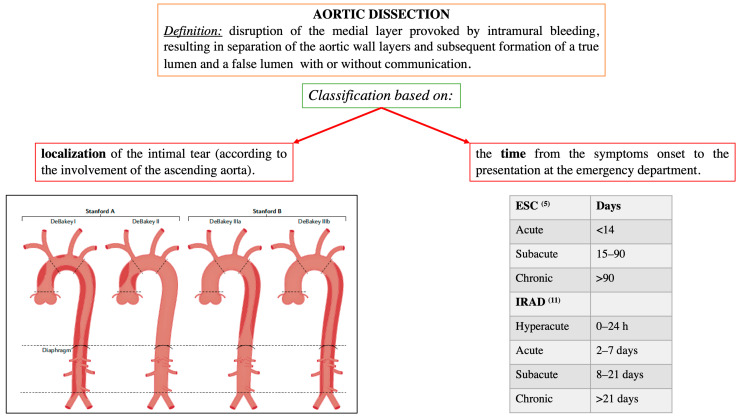
Localization and time-based definitions of aortic dissection, according to the European and American guidelines [5,6], and the International Registry of Acute Aortic Dissection (IRAD) [11]. Modified by Bossone et al., Nat Rev Cardiol. 2021 May;18(5):331–348 [10].

**Figure 2 jpm-12-01148-f002:**
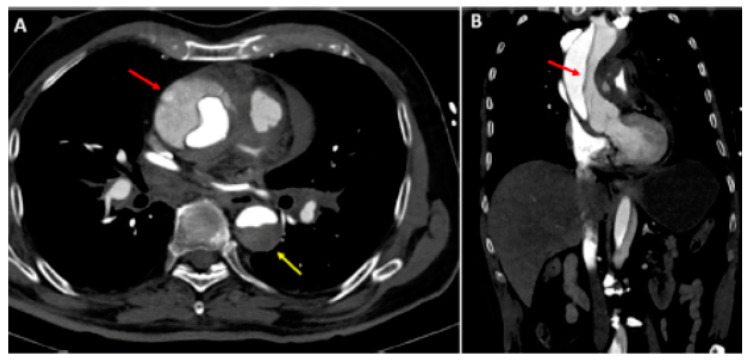
Type A aortic dissection: (**A**) Computed tomography axial view. Ascending aorta (red arrow) and descending aorta (yellow arrow) dissection; (**B**) dissection extending into the right subclavian artery (sagittal view). Reprinted/adapted with permission from Ref. [18]. Copyright year 2021, with permission from Elsevier.

**Figure 3 jpm-12-01148-f003:**
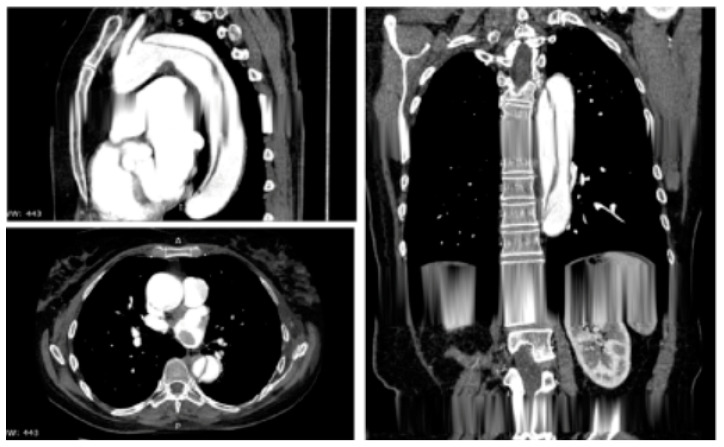
Contrast-enhanced computed tomography scan reconstruction of type B acute aortic dissection. Reprinted/adapted with permission from Ref. [12]. Copyright year 2020, with permission from Elsevier.

**Figure 4 jpm-12-01148-f004:**
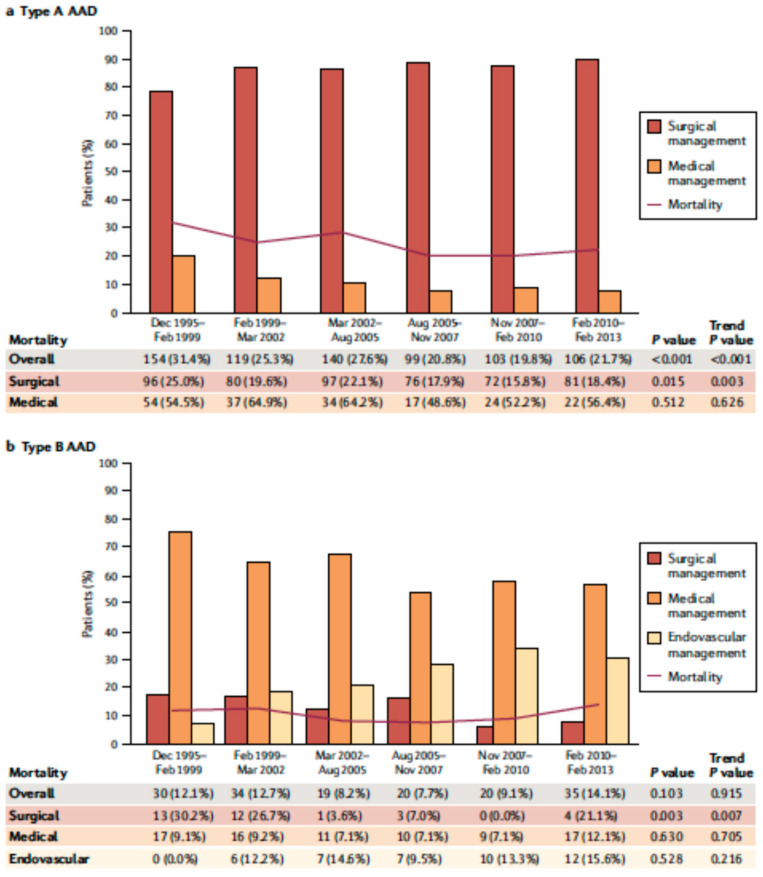
Over time type A (**a**) and type B (**b**) acute aortic dissection in-hospital mortality. Modified by Bossone et al., Nat Rev Cardiol. 2021 May;18(5): 331–348 [10].

**Table 1 jpm-12-01148-t001:** Clinical presentation of AAD, according to type (A, B and A + B) of the major population-based studies.

Study	AAD, n	Women, n (%)	Age (Mean, ± SD)			Clinical Presentation											
						Shock, n (%)			Pericardial Effusion/Tamponade, n (%)			Neurological Disorders, n (%)			Malperfusion, n (%)		
			Women	Men	*p*-Value	Women	Men	*p*-Value	Women	Men	*p*-Value	Women	Men	*p*-Value	Women	Men	*p*-Value
**TYPE A Add**																	
Fukui et al. [19] 2006–2013	504	245 (48.6)	71.5 ± 9.8	59.7 ± 13.4	<0.001	68 (27.8)	53 (20.5)	0.07	52 (22.4)	40 (15.4)	0.05	17 (6.9)	11 (4.3)	0.26	-	-	-
Chemtob et al. [20] 2005–2014	1154	373(32)	65 ± 11	60 ± 12	<0.001	91 (26.1)	161 (22.3)	0.19	70 (19.2)	120 (15.7)	0.17	-	-	-	-	-	-
Conway et al. [21] 2000–2010	251	79 (31.4)	67 (20–87)	58 (19–83)	<0.001	9 (12)	21 (13)	0.78	-	-	-	-	-	-	-	-	-
Friederich et al. [22] 2001–2016	368	126 (34.2)	67.5 ± 11.8	60.4 ± 12	<0.001	10 (7.9)	17 (7.2)	0.79	-	-	-	-	-	-	-	-	-
Rilsky et al. [23] 2006–2015	3380	1234 (37)	65.5 ± 12.7	59.2 ± 13.3	<0.001	-	-	-	270 (21.9)	430 (20)	0.2	29 (2.4)	83 (3.9)	0.018	425 (32.8)	827 (38.5)	0.001
Sabashnikov et al. [24] 2006–2015	240	87 (36.3)	68 58–76 (Median IQR)	59 50–71 (Median IQR)	<0.001	17 (23.9)	20 (28.2)	0.56	32 (45.1)	33 (46.5)	0.866	13 (18.3)	12 (16.9)	0.94	23 (32.4)	18 (25.4)	0.35
Suzuki et al. [25] 2004–2016	303	147 (48.5)	72.6 ± 10.3	63.0 ± 12.9	<0.001	51 (35)	55 (35)	0.91	41 (28)	47 (30)	0.67	23 (16)	18 (12)	0.29	34 (23)	40 (26)	0.61
**TYPE B AAD**																	
Liang et al. [29] 2009–2012	9855	4293 (43.6)	69.8 ± 15.5	62.8 ± 15.6	<0.001	185 (4.3)	274 (4.9)	0.2	-	-	-	38 (0.9)	96 (1.7)	<0.001	-	-	-
Takahashi et al. [30] 2013–2017	2372	695 (29.3)	76 66–84 (Median IQR)	68 57–77 (Median IQR)	<0.001	20 (2.9)	43 (2.6)	0.6	-	-	-	-	-	-	14 (2)	57 (3.4)	0.085
**TYPE A + B AAD**																	
Liu et al. [27] 2002–2016	884	221 (24.9)	51.4 ± 11.8	55.1 ± 12.5	0.10	-	-	-	47 (22.3)	112 (16.6)	0.06	23 (10.9)	50 (7.4)	0.11	7 (3.3)	47 (7)	0.05
Maitusong et al. [26] 2010–2015	400	96 (24)	54.2 ± 12.4	49.6 ± 12.6	<0.001	1 (1)	11 (3.6)	0.31	12 (12.5)	15 (4.9)	<0.001	2 (2.1)	27 (8.9)	0.04	4 (4.2)	36 (11.8)	0.03

**Table 2 jpm-12-01148-t002:** Short- and long-term outcomes of AAD, according to type (A, B and A+B) of the major population-based studies.

Study	AAD, n	Women, n (%)	In-Hospital Mortality, n (%)				30 Days-Mortality, n (%)			5-Years Survival, %			10-Years Survival, %		
				Women	Men	*p*-Value	Women	Men	*p*-Value	Women	Men	*p*-Value	Women	Men	*p*-Value
**TYPE A AAD**															
Fukui et al. [19] 2006–2013	504	245 (48.6)	Surgical treated	-	-	-	11 (4.5)	15 (5.8)	0.64	80.1	89.3	0.06	-	-	-
Chemtob et al. [20] 2005–2014	1154	373(32)	Surgical treated	34 (9.1)	52 (6.7)	0.17	65 (19.2)	138 (18.9)	0.99	-	-	-	-	-	-
Conway et al. [21] 2000–2010	251	79 (31.4)	Surgical treated	15 (19)	29 (17)	0.69	-	-	-	-	-	-	58	63	0.28
Friederich et al. [22] 2001–2016	368	126 (34.2)	Surgical treated	23(18.4)	38 (16.2)	0.60	24 (19)	40 (16.5)	0.54	70	71	0.5	-	-	-
Rilsky et al. [23] 2006–2015	3380	1234 (37)	Surgical treated	-	-	-	202 (16.3)	356 (16.6)	0.17	-	-	-	-	-	-
Sabashnikov et al. [24] 2006–2015	240	87 (36.3)	Surgical treated	6 (7)	8 (5.6)	0.99	18 (21)	21 (14)	0.27	62.9	62.8	0.25	-	-	-
Suzuki et al. [25] 2004–2016	303	147 (48.5)	Surgical treated	13 (9)	16 (10)	0.68	12 (8)	14 (9)	0.80	-	-	-	59	65.7	0.81
Liu et al. [27] 2002–2016	355	97 (27.3)	Surgical treated	10 (5.1)	30 (4.7)	0.82	-	-	-	-	-	-	-	-	-
Maitusong et al. [26] 2010–2015	154	52 (34.3)	Overall (surgical + medical) Surgical treatment Medical treatment	25 (75.8) 4 (40) 21 (91.3)	55 (44.6) 4 (11.8) 51 (58.6)	<0.01 0.04 <0.01	-	-	-	-	-	-	-	-	-
**TYPE B AAD**															
Liang et al. [29] 2009–2012	9855	4293 (43.6)	Overall (TEVAR + surgical + medical)	498 (11.6)	594 (10.7)	0.2	-	-	-	-	-	-	-	-	-
Takahashi et al. [30] 2013–2017	2372	695 (29.3)	Overall Medically managed Interventionally treated Endovascularly treated Surgically treated	37(5.3) 26 (4.1) 11 (17.5) 4 (14.8) 7 (19.4)	46 (2.7) 26 (1.8) 20 (8.7) 8 (7.6) 12 (9.6)	0.002 0.002 0.04 0.26 0.10	-	-	-	-	-	-	-	-	-
Liu et al. [27] 2002–2016	533	114	Overall (surgical + TEVAR + medical managed)	1 (0.5)	8 (1.3)	0.63	-	-	-	-	-	-	-	-	-
Maitusong et al. [26] 2010–2015	400	96 (24)	Overall Medically managed Surgically treated	5 (7.9) 3 (12) 2 (5.1)	7 (3.8) 2 5 (10.2) 2 (1.5)	0.19 0.77 0.22	-	-	-	-	-	-	-	-	-
**TYPE A + B AAD**															
Maitusong et al. [26] 2010–2015	400	96 (24)	Overall	30 (31.3)	62 (20.4)	0.02	-	-	-	-	-	-	-	-	-

**Table 3 jpm-12-01148-t003:** AAD gender-related differences in epidemiological, clinical, treatment and outcomes from International Registry of Acute Aortic Dissection (IRAD) [11,32].

	IRAD 2004 [11] n = 1078 (Female = 346; Male = 732) TYPE A + B AAD			IRAD-IVC 2021 [32] n = 2083 (Female = 969; Male = 1854) TYPE A AAD Treated with Endovascular, Surgical, or Hybrid Procedures for Aortic Dissection		
	*Female*	*Male*	*p-Value*	*Female*	*Male*	*p-Value*
**Age mean ± SD**	66.7 ± 13.9	60.3 ± 13.7	<0.001	65.4 ± 13.4	58.6 ± 13.3	<0.001
**Risk factors/medical history**						
Hypertension, n (%)	258 (77.5)	497 (69.2)	0.006	725 (80.5)	1286(74.4)	<0.001
Diabetes, n (%)	19 (5.8)	32 (4.5)	0.40	113 (13.1)	147 (8.9)	0.001
Current smoker, n (%)	--	--	--	177 (28.1)	425 (33.4)	0.017
Ever smoker, n (%)	--	--	--	303 (48)	706 (55.5)	0.002
Bicuspid aortic valve, n (%)	--	--	--	17 (2.0)	72 (4.4)	0.003
Previous cardiac surgery, n (%)	50 (16.9)	142 (24.0)	0.02	85 (10)	210 (12.7)	0.042
Coronary artery bypass, n (%)	12 (3.7)	45 (6.6)	0.07	21 (2.5)	87 (5.3)	0.001
Aortic aneurysm/dissection repair, n (%)	20 (6.2)	70 (10.1)	0.04	34 (4.0)	90 (5.5)	0.116
**Signs/Symptoms at onset**						
Abrupt onset of pain, n (%)	266 (82.6)	629 (89)	0.004	--	--	--
Altered consciousness, n (%)	43 (13.0)	63 (9.0)	0.05	111 (11.5)	139 (7.5)	0.001
Congestive heart failure, n (%)	28 (9.1)	36 (5.4)	0.03	--	--	--
Any pulse deficit, n (%)	57 (19.2)	202 (31.7)	<0.001	--	--	--
Malperfusion with shock, n (%)	34 (10.5)	69 (9.8)	0.71	303(31.3)	411 (22.2)	<0.001
**Diagnostic imaging findings**						
Intramural hematoma, n (%)	40 (11.7)	74 (10.2)	0.49	188 (19.4)	244 (13.2)	<0.001
Periaortic hematoma, n (%)	86 (27.6)	141 (21.4)	0.03	115 (24.8)	174 (18.6)	0.007
Complete FL thrombosis, n (%)	45 (17.0)	58 (10.7)	0.01	61 (17.2)	77 (10.2)	0.001
Partial FL thrombosis, n (%)	76 (28.8)	136 (25.2)	0.28	88 (24.8)	146 (19.4)	0.039
Patent FL, n (%)	--	--	--	206 (58)	531 (70.4)	<0.001
Coronary artery compromise n (%)	30 (10.8)	39 (6.9)	0.05	51 (12.3)	92 (11.3)	0.584
Pericardial effusion, n (%)	128 (38.6)	195 (28.6)	0.001	290 (49.5)	468 (39.8)	0.001
AR more than mild, n (%)	--	--	--	84 (53.5)	216 (64.9)	0.016
Distal extent (iliofemoral), n (%)	--	--	--	80 (12.5)	220 (16.9)	0.015
**Electrocardiogram**						
Low voltage (ECG), n (%)	19 (6.3)	21 (3.3)	0.03	--	--	--
**Treatment (type A + B)**						
Surgery, n (%)	173 (50.0)	448 (61.2)	0.001	--	--	--
Endovascular, n (%)	15 (4.3)	28 (3.8)	0.69	--	--	--
Medical, n (%)	256 (35.0)	158 (45.7)	0.001	--	--	--
Initial management with BB, n (%)	179 (55.6)	430 (62.1)	0.05	--	--	--
**Treatment type A**						
Surgery, n (%)	161 (70.6)	388 (86.8)	<0.001	--	--	--
Endovascular, n (%)	2 (0.9)	1 (0.2)	0.27	--	--	--
Medical, n (%)	65 (28.5)	58 (13.0)	<0.001	--	--	--
Bentall, n (%)	--	--	--	129 (22.8)	391 (32.4)	<0.001
Complete arch replacement, n (%)	13 (7.7)	51 (11.7)	0.15	119 (15.2)	315 (20.6)	0.002
Elephant trunk, n (%)	--	--	--	24 (3.3)	92 (6.5)	0.002
Aortic valve replacement, n (%)	--	--	--	200 (26.6)	507 (34.5)	<0.001
Cardiopulmonary bypass time, min (IQR)	--	--	--	182 (145–234)	201 (157–248)	<0.001
**Treatment type B**						
Surgery, n (%)	12 (10.2)	60 (21.1)	0.009	--	--	--
Endovascular, n (%)	13 (11.0)	27 (9.5)	0.64	--	--	--
Medical, n (%)	93 (78.8)	198 (69.5)	0.06	--	--	--
**Post operative complications**						
Acute renal failure, n (%)	53 (16.9)	128 (18.7)	0.49	164 (17.7)	393 (21.2)	0.029
Hypotension, n (%)	107 (34.1)	164 (23.9)	0.001	--	--	--
Cardiac tamponade, n (%)	52 (16.5)	71 (10.5)	0.007	--	--	--
Limb ischemia, n (%)	23 (7.4)	80 (11.8)	0.04	--	--	--
**In-hospital overall mortality (type A + B)**	104 (30.1)	154 (21.0)	0.001	--	--	--
**In-hospital mortality type A (overall)**	87 (38.2)	119 (26.6)	0.002	--	--	--
**In-hospital mortality (surgical treatment), n (%)**	52 (31.9)	85 (21.9)	0.013	162 (16.7)	256 (13.8)	0.039
**In-hospital mortality (non-surgical treatment), n (%)**	35 (53.8)	34 (58.6)	0.59	--	--	--
**In-** **hospital mortality type B (overall)**	17 (14.4)	35 (12.3)	0.56			
**In-hospital mortality (surgical treatment), n (%)**	5 (20.0)	19 (21.8)	0.84	--	--	--
**In-hospital mortality (non-surgical treatment), n (%)**	16 (8.1)	12 (12.9)	0.19	--	--	--
**5-years survival, n (%)**	--	--	--	800 (82.6)	1592 (85.9)	0.136

FL = False lumen; AR = aortic regurgitation; BB = beta-blockers.

## Data Availability

Not applicable.

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
