# Peer review of "Gender Differences in Acute Aortic Dissection"

_jpm, 2022, doi:10.3390/jpm12071148_

Round 1
Reviewer 1 Report
I read the manuscript titled "Gender differences in acute aortic dissection" with great interest. Here are my comments:
1. A table with presentation and outcomes difference between two genders would be helpful in addition to the details in manuscript
2. I encourage the authors to add findings of ew study published in JAHA titled "Sex-related differences in clinical features and in-hospital outcomes of type B acute aortic dissection" by Takahashi et al
Author Response
I read the manuscript titled "Gender differences in acute aortic dissection" with great interest. Here are my comments:
- A table with presentation and outcomes difference between two genders would be helpful in addition to the details in manuscript
We are grateful for your comments. We have added presentation and outcomes tables (Tables 1 and 2 marked in red).
- I encourage the authors to add findings of few study published in JAHA titled "Sex-related differences in clinical features and in-hospital outcomes of type B acute aortic dissection" by Takahashi et al.
Thank so much for your suggestion. We added Takahashi et al in the text (Type B AAD, Page 5, Line 29, in red) and in the references.
Please see the manuscript attached

Reviewer 2 Report
the manuscript is overall timely and well written. Nevertheless, there are some aspects which might benefit from Authors’ reworking to strengthen the message of the study. The introduction paragraph might be significantly shortened, besides, there are several self-citations which add are theoretically correct but add very little to the sake of clarity and to the main aim of the review. The Authors decided to separate the IRAD experience from the other published evidence (including evidence from other relevant databases), maybe this ended up in a less straightforward exposure and, to some instances, to a duplicate message. Finally, the conclusion paragraph should better characterize the clinical bottom line of the presented evidence and should comment on how these data might impact on the personalized treatment of AAD.
Author Response
The manuscript is overall timely and well written. Nevertheless, there are some aspects which might benefit from Authors’ reworking to strengthen the message of the study. The introduction paragraph might be significantly shortened, besides, there are several self-citations which add are theoretically correct but add very little to the sake of clarity and to the main aim of the review. The Authors decided to separate the IRAD experience from the other published evidence (including evidence from other relevant databases), maybe this ended up in a less straightforward exposure and, to some instances, to a duplicate message. Finally, the conclusion paragraph should better characterize the clinical bottom line of the presented evidence and should comment on how these data might impact on the personalized treatment of AAD.
- Thanks for your comments.
- As suggested, we have shortened the introduction (and eliminating some self-citations) and improved the conclusions (Page 14 Line 246-248 in red).
We have chosen to highlight the IRAD experience separately, as it represents by far the largest and the most comprehensive international registry.
Please see the manuscript attached

Round 2
Reviewer 2 Report
suitable for publication in the present form